# The Pathogen Profile of a Honey Bee Queen Does Not Reflect That of Her Workers

**DOI:** 10.3390/insects11060382

**Published:** 2020-06-20

**Authors:** Jessica L. Kevill, Katie Lee, Michael Goblirsch, Erin McDermott, David R. Tarpy, Marla Spivak, Declan C. Schroeder

**Affiliations:** 1Department of Veterinary Population Medicine, University of Minnesota, 1365 Gortner Ave., St Paul, MN 55108, USA; 2Department of Entomology, University of Minnesota, 1980 Folwell Ave, Suite 219, St Paul, MN 55108, USA; leex1444@umn.edu (K.L.); michael.goblirsch@usda.gov (M.G.); spiva001@umn.edu (M.S.); 3United States Department of Agriculture, Agricultural Research Service, Southeastern Area, Thad Cochran Southern Horticultural Research Laboratory, 810 Highway 26 W., Poplarville, MS 39470, USA; 4Department of Entomology & Plant Pathology, North Carolina State University, Raleigh, NC 27695, USA; eemcderm@ncsu.edu (E.M.); drtarpy@ncsu.edu (D.R.T.); 5School of Biological Sciences, University of Reading, Reading RG6 6LA, UK

**Keywords:** honey bees, queen bees, horizontal transmission, LSV, DWV, Nosema, pathogens

## Abstract

Throughout a honey bee queen’s lifetime, she is tended to by her worker daughters, who feed and groom her. Such interactions provide possible horizontal transmission routes for pathogens from the workers to the queen, and as such a queen’s pathogen profile may be representative of the workers within a colony. To explore this further, we investigated known honey bee pathogen co-occurrence, as well as pathogen transmission from workers to queens. Queens from 42 colonies were removed from their source hives and exchanged into a second, unrelated foster colony. Worker samples were taken from the source colony on the day of queen exchange and the queens were collected 24 days after introduction. All samples were screened for Nosema spp., Trypanosome spp., acute bee paralysis virus (ABPV), black queen cell virus (BQCV), chronic bee paralysis virus (CBPV), Israeli acute paralysis virus (IAPV), Lake Sinai virus (LSV), and deformed wing virus master variants (DWV-A, B, and C) using RT-qPCR. The data show that LSV, Nosema, and DWV-B were the most abundant pathogens in colonies. All workers (*n* = 42) were LSV-positive, 88% were Nosema-positive, whilst pathogen loads were low (<1 × 10^6^ genome equivalents per pooled worker sample). All queens (*n* = 39) were negative for both LSV and Nosema. We found no evidence of DWV transmission occurring from worker to queen when comparing queens to foster colonies, despite DWV being present in both queens and workers. Honey bee pathogen presence and diversity in queens cannot be revealed from screening workers, nor were pathogens successfully transmitted to the queen.

## 1. Introduction

Honey bees are highly social organisms. A typical healthy colony is headed by a single egg-laying queen, more than 20,000 female workers, and several hundred male drones. Worker bees perform a number of altruistic tasks to ensure the survival of the colony. These tasks involve foraging for food, defending the colony, nursing offspring, cleaning, and attending to the queen [1]. The attendant bees that groom and feed the queen are typically aged between 1 and 52 days [2]. Queen bees are fed solely with royal jelly, which is a glandular secretion produced by attendant workers younger than 23 days old [2]. Royal jelly is also fed to worker larvae by nurse bees in the first 3 days of development, after which their diet is switched to a pollen–honey mixture known as beebread. These social interactions and the fact that colonies are densely populated provide the perfect opportunity for the intracolony transmission of pathogens [3].

Pathogen transmission can occur horizontally or vertically. Vertical transmission occurs when pathogens are transmitted from parent to offspring and are often associated with less virulent pathogens, as the next generation of host is required for future transmission [4,5,6,7,8,9]. Horizontal transmission can be either direct or indirect, with each route having differing degrees of virulence. Direct transmission occurs when a host comes into contact with infectious material [8]. Indirect transmission requires a vector to deliver the pathogen to the host, such as the feeding activities of the parasitic mite *Varroa destructor* (hereafter referred to as Varroa) [10,11]. Indirect horizontal transmission of viral pathogens in the presence of Varroa has been shown to cause lethal infections in honey bee colonies [12,13,14,15].

There are a number of pathogens that can be transmitted horizontally within a honey bee colony. For the most part, these are positive-sense RNA viruses that are often associated with Varroa infestation [16,17,18]. Approximately 23 viruses have been identified so far in honey bees [18,19,20]. Multiple viral infections can be present in colonies and individual honey bees at any one time [21,22]. The most commonly detected viruses in honey bees are the Dicistroviridae Israeli acute paralysis virus (IAPV), Kashmir bee virus (KBV), acute bee paralysis virus (ABPV), black queen cell virus (BQCV); the Iflaviridae deformed wing virus variants (DWV-A, DWV-B, DWV-C), Kakugo virus (KV), sacbrood virus (SBV), slow bee paralysis virus (SBPV), recently classified Sinai viruses (Lake Sinai virus (LSV1-7)), and the unclassified chronic bee paralysis virus (CBPV) [23].

Viruses are present during every life stage of honey bees, which is particularly true for DWV [9]. Common honey bee viruses can be transmitted horizontally during trophallaxis in their salivary secretions and whilst performing colony-related tasks [24]. The oral transmission of viruses is an important mechanism for maintaining viral infections in the absence of a vector [25]. Locke et al. (2017) [26] found that subclinical DWV viral loads were maintained in colonies after Varroa management. Viruses are present in food stores; DWV and BQCV have both been detected in honey and pollen [27]; ABPV, CBPV, KBV, and SBV have been detected only in pollen [8,24], and KBV and SBV have been found in royal jelly [28]. The horizontal transmission of viruses to workers in food has been shown to influence vertical transmission from queen to offspring. In Singh et al. (2010) [27], queens that were verified as virus-free had their colonies fed DWV- and SBV-positive honey and beebread. Queens began to lay DWV-positive eggs after 1 week of worker exposure to DWV-positive beebread and 2 weeks exposure to DWV-infected honey. CBPV has also been evidenced to be transmitted to queens from workers in cage experiments [29]. A number of viruses have been shown to be present in the hypopharyngeal glands of nurse bees, such as SBV [28], ABPV [30], and DWV [31]. Many of these viruses have also been detected in the honey bee gut and feces, highlighting the importance of food borne transmission [8,30,32,33]. Therefore, queens are susceptible to acquiring viruses from attending workers.

There are several other non-viral pathogens detected in honey bees. These include the bacteria that cause European foulbrood (*Melissococcus plutonius*) [34,35] and American foulbrood (*Paenibacillus larvae*) [36,37], as well as the fungus that causes chalkbrood (*Ascosphaera apis*) [38]. These pathogens cause larval death during development, leading them to be termed brood diseases. Obligate parasites such as *Nosema apis*, *N. ceranae,* and Trypanosome spp. are also found in honey bees. Nosema are unicellular parasites belonging to the order Microsporidia, while Trypanosomes are a kinetoplastid. Nosema and Trypanosomes infect the digestive system of honey bees, and as such are transmitted horizontally via the fecal–oral route, while Nosema has also been proven to be transmitted sexually to queens from drones [39]. Research shows that *N. ceranae* is dominant in US honey bee populations but infection has not been correlated to colony losses [40], unlike what has been reported in other countries [41,42]. Trypanosomes are also present in US honey bee colonies, but they are not as common as Nosema or viral pathogens [43].

Pathogens are present in most aspects of colony life, including the incredibly divergent life histories between castes. Queens are long-lived compared to workers, and as such they are considered to be less susceptible to infection [44]. Queens historically live on average for 2–3 years [45], although likely less in recent years [46]. In contrast, most workers during the active season live 3–6 weeks [47], whilst the longest living workers are overwintering bees that can survive for several months [48]. During the active season, worker bees within a colony are constantly exposed to pathogens via their interactions with each other and their environment. The queen is surrounded by nurse bees that tend to her and likely expose her to the same sources of infection, yet she is likely to survive. Therefore, it is hypothesized that if queens are constantly exposed to pathogens within the colony by attending worker bees and these pathogens are transmitted to the queen, then the pathogen profiles of the queens will match that of the workers. To address the horizontal transmission of pathogens from worker to queen, we conducted a queen exchange experiment. The pathogen profiles of exchanged queens, the workers in the colonies from which they originated (source), and the workers in the colonies from which queens were exchanged (foster) were quantified and compared. Overall, our results show that the queen pathogen profile does not match that of the adult workers in the source or foster colonies.

## 2. Methods

### 2.1. Queen Exchange and Sample Collection

Adult honey bee workers and queens were collected from 42 commercial honey bee colonies, located across four operations in either Minnesota or Texas during 2017. All colonies were headed by queens <6 months old and were not selected based on their sealed brood pattern as part of the Lee et al. (2019) [49] study. No association between brood pattern and pathogen presence was made by Lee et al. (2019); however, the samples provide the opportunity to assess intracolony pathogen transmission. Experimental colonies were identified during the months of May and June [49] to ensure that all queens were young and comparable in age. Queens were exchanged into foster colonies located in the same or adjacent apiary. Prior to exchange, queens were marked and temporarily caged in a California Mini Queen Cage (C.F. Koehnen and Sons, Inc. in Glenn, CA, USA). The queens were placed into their corresponding foster colony and released manually 3 days later. On the day of initial queen introduction, a sample of approximately 50 adult workers was collected from a brood frame in the colonies, and as such likely containing a large number of newly emerged and nurse bees [50]. Exchanged queens were removed from foster colonies 24 days after their release from the cages [49], as 24 days allowed queens to become established and tended to by workers in the foster colony. Sampling and queen exchange were conducted as follows: The queens from colony one and two were removed and the workers from both colonies were sampled. The queen from colony one was placed into colony two, the queen from colony two was placed into colony one, and so on. Therefore, the workers from colony one served as the source workers for queen one and the foster workers for queen two when comparing queens to source and foster colonies. This was repeated for all 42 colonies; no one colony used in this experiment kept its original queen. Three of the queens were not sampled, as the queens died. This resulted in 81 samples, with a total 39 queens and 42 pooled samples of 50 adult workers.

Varroa levels were established using an alcohol wash [51] to dislodge mites from adult bees, as part of the Lee et al. (2019) study. Overall mite loads were low in colonies at <3 mites/100 bees. As such, Varroa pathogen transmission was not considered within this study.

### 2.2. Sample Processing

Spermathecae were dissected and removed from all queens as part of the Lee et al. (2019) study [49]; therefore, homogenization was performed on the remaining individual queen tissues and separate pools of 50 adult workers. Samples were homogenized in an appropriate volume of TRIzol (Thermo Fisher Scientific, HQ in Waltham, MA, USA). Total RNA was extracted using the standard phenol–chloroform protocol [52]. RNA concentration and quality were determined using a NanoDrop spectrophotometer.

### 2.3. Molecular Analysis

RNA concentration was diluted to 200 ng/µL before cDNA synthesis with the BioBasic Reverse Transcriptase Mix (Biobasic Inc. in Markham, ON, Canada). Quantitative PCR (qPCR) was performed following the methods of Lee et al. (2019) for the detection of the following pathogens: acute bee paralysis virus (ABPV), black queen cell virus (BQCV), chronic bee paralysis virus (CBPV), Israeli acute paralysis virus (IAPV), and Lake Sinai virus (LSV). Nosema and Trypanosome (Tryp) levels were quantified in samples by quantitative PCR using universal primers designed to detect both Nosema species (*N. ceranae* and *N. apis*) and to universally detect Tryp. species. Primers for these targets were designed to amplify in the small subunit ribosomal RNA (SSU rRNA) in order to use the same RNA extraction, as was used for the viral targets. All primers can be found in the work of Alburaki et al. (2018) [53], except the Tryp. primer set, which was as follows: (forward)-GAGTGTGGCAGGACTACCC, (reverse)-TGCACCAACCACGAAATGA. RT-qPCR was performed in triplicate with Power-Up^™^ SYBR^®^ Green Mastermix (Thermo Fisher Scientific, HQ in Waltham, MA, USA) on a 384-well QuantStudio Flex 6 instrument (Thermo Fisher Scientific, HQ in Waltham, MA, USA). Cycling conditions were adapted from Power-Up^™^ SYBR^®^ Green protocols. The standard curve for copy number quantification was determined by running a dilution series of known plasmid DNA standards on each plate, as per Lee et al. (2019).

DWV master variants (DWV-A, B, and C) were detected and quantified using a previously described reverse transcriptase qPCR (RT-qPCR) method [54]. Briefly, total RNA was diluted to 50 ng/µL and RT-qPCR was performed in triplicate using Quant Studio 3 (Applied biosystems, USA) and a Power-Up^™^ SYBR^®^ Green RNA-to-Ct 1-Step Kit (Applied Biosystems, Foster City, CA, USA). Each reaction contained 1 µL total RNA (50 ng/µL), 0.08 µL reverse transcriptase, 1 µL reverse primer (DWV-A, B, or C), 1 µL universal forward primer, and 1.92 µL molecular grade water. Reverse transcription occurred at 45 °C for 10 min and denaturation occurred at 95 °C for 10 min, followed by 35 cycles of denaturation at 95 °C for 15 s, annealing at 58 °C for 15 s (DWV-A and DWV-B) or 61 °C (DVW-C), and extension at 72 °C for 15 s. A high-resolution melt analysis was performed between 72 °C and 90 °C, at 0.1 °C increments, each with a 5 s hold period. The standard curve for copy number quantification was determined by running a dilution series of known plasmid RNA standards on each plate.

### 2.4. Analysis

Pathogen prevalence was calculated for all data by calculating the percentage of positive and negative samples per pathogen, regardless of the sample type (queen or worker). Pathogen prevalence was then calculated for queens and workers separately. Prevalence was determined independently of exchange to assess the overall population of pathogens and if pathogens were specific to pooled workers or queens. 

Copy number (pathogen load) per queen or pooled worker sample for the pathogens screened were calculated as per Lee et al. (2019) [49], except for DWV, which was quantified as per Kevill et al. (2017) [54]. Pathogen detection thresholds for all the pathogens screened were set at 30 Ct values (100 copies), as beyond this threshold samples cannot be reliably quantified. Colonies were negative for CBPV and DWV-C; therefore, analysis was conducted on ABPV, BQCV, DWV-A, DWV-B, IAPV, LSV, Trypanosome spp., and Nosema spp.

A Kruskal–Wallis test was used to compare DWV-A and DWV-B viral load in pooled workers and exchanged queens. Post-hoc analysis was conducted using a Dunn’s test of multiple pairwise comparisons [55]. The significance threshold was set at α = 0.05 for all tests, and when multiple comparisons were made the significance level was adjusted depending on the number of comparisons made using a Hochberg’s step-up method for Bonferroni correction (significance threshold/number of comparisons).

## 3. Results

### 3.1. Pathogen Prevalence

Pathogen prevalence varied in the 81 samples. LSV was the most commonly detected pathogen, followed by Nosema and DWV-B, with 52%, 46%, and 42% of samples testing positive, respectively. DWV-A and BQCV was detected in 16% and 15% of the samples, respectively. ABPV, IAPV, and Trypanosome were found in <10% of samples (Figure 1). All samples were negative for CPBV and DWV-C.

### 3.2. A Comparison of Pathogens in Pooled Workers and Queens

Regardless of exchange, the prevalence of pathogens by sample type (queen or pooled workers) shows that the pooled worker samples from all 42 colonies were positive for LSV, 88% were positive for Nosema, 29% were positive for BQCV, 14% were positive for Trypanosome, and 7% were positive for ABPV. Only 12% of pooled workers and 3% (*n* = 1) of queens were IAPV-positive, which was too infrequent to conduct a suitable comparison. None of the queens were positive for the pathogens ABPV, BQCV, LSV, Trypanosome, or Nosema. DWV-A and DWV-B were detected in both pooled workers and queens. DWV-A was present in 19% of pooled workers and 13% of queens. DWV-B was more prevalent than DWV-A, as 50% of pooled workers and 33% of queens were DWV-B-positive (Figure 2).

Both DWV-A and B variants were detected in higher viral loads compared to the other pathogens (ABPV, BQCV, IAPV, LSV, Trypanosomes, and Nosema; Figure 3). DWV-A was detected at the order of 10^4^–10^7^ genome equivalents, while DWV-B was detected at 10^6^–10^9^ for the majority of samples. IAPV, LSV, Trypanosome, and Nosema were detected in pooled worker bees only (with the exception of one IAPV-positive queen) at viral loads <10^6^ (Figure 3). Multicomparisons revealed significant differences in DWV variants and loads in pooled workers and queens (Kruskal–Wallis test; *n* = 47, H 14.06, *df* 3, *p* < 0.05), regardless of exchange. When comparing DWV-A and DWV-B per sample type (queen or worker), DWV-B was always present at a higher load than DWV-A (Dunn’s test, *n* = 47; queens *p* < 0.05, workers *p* < 0.005). Comparisons of DWV-A and DWV-B revealed that the mean viral loads between queens and pooled workers were not significant (Dunn’s test, *n* = 47, *p* = 0.30 and *p* = 0.40, respectively).

### 3.3. DWV Transmission between Pooled Workers and Queens

The majority of queens and corresponding source and foster colonies were negative for DWV-A, which was detected in only five of the 39 queens analyzed and eight of the 42 pooled worker samples. There were no situations where queens tested postive after 24 day exposure to DWV-A-positive workers from a foster colony. In contrast, on four occasions we observed that DWV-A-positive queens came from a source colony with DWV-A-positive workers (colony IDs 6, 10, 11, 13; Figure 4A). The DWV-A viral loads in three of these four queens were one-fold higher than the correspoding workers in the source colony ( colony IDs 6, 11, 13), while one queen had a similar load to its workers source colony (colony ID 10, Figure 4A). Surprisingly, one queen was DWV-A-positive despite the corresponding source and foster colony workers testing negative (colony ID 5; Figure 4A). Six queens from DWV-A-negative source colonies, when exchanged into a DWV-A-positive worker foster colony, remained DWV-A-free (colony IDs 1, 14, 26–28, 31; Figure 4A). Interestingly, four queens that tested DWV-A-negative came from source colonies with DWV-A-positive workers (colony IDs 7, 9, 12, 21; Figure 4A).

DWV-B was detected in 13 of the 39 queens and 21 of the 42 pooled worker samples. Of these 13 queens, nine were from source colonies with DWV-B-positive workers (colony IDs 5–13; Figure 4B), two were exchanged into a DWV-B-positive worker colony (colony IDs 1 and 2; Figure 4B), and two were positive despite both the source and foster colony workers being negative (colony IDs 3 and 4; Figure 4B). Out of the 26 negative queens, six were exchanged into DWV-B-positive worker foster colonies (colony IDs 26–31; Figure 4B). In addition, six queens from DWV-B-positive worker source colonies exchanged into DWV-B-positive worker foster colonies were DWV-B-negative (colony IDs 14–19; Figure 5). Furthermore, six DWV-B-negative queens from positive worker source colonies were also identified (colony IDs 20–25; Figure 4B). DWV-B viral loads in positive queens followed the same trend as DWV-A, i.e., DWV-B was detected at higher loads than the workers in the majority queens (colony IDs 1–6, 8, 10, 11, 13; Figure 4B). Two DWV-B-positive queens had comparable DWV loads to those of the source colony workers (colony IDs 9 and 12; Figure 4B).

Co-infections of DWV-A and DWV-B were detected in all DWV-A-positive queens and pooled worker samples (Figure 4). Queens were negative for the remaining pathogens, with the exception of one IAPV-positive queen (colony IDs 36; Figure 5). IAPV infection was not detected in the source (colony IDs 36) or foster colony (colony IDs 33) for the IAPV-positive queen (Figure 5). Workers often had both Nosema and LSV infection (excluding colony IDs 4, 9, 29, 31, 40; Figure 5), as these were the most prevalent pathogens. Co-infections with multiple pathogens were common, with 90% (*n* = 38) of pooled worker samples positive for two or more pathogens. Interestingly, 33% (*n* = 14), 21% (*n* = 9), 5% (*n* = 2) and 10% (*n* = 4) of pooled worker samples were infected with three, four, five, and six different pathogens, respectively.

## 4. Discussion

The results of this study reveal that the pathogen profile of a newly introduced queen does not reflect that of the resident workers. Furthermore, the 24 day exposure period of queens to workers positive for at least two pathogens did not result in detectable pathogen loads in queens, nor did the queens acquire pathogens from source colonies, leading to the conclusion that adult queens may not readily acquire pathogens from workers. If the transmission of pathogens did take place, then infection did not take hold, or alternatively the queens developed a resistance to the pathogens screened. Age-related immunity may play a part in lack of pathogen transmission, as Chaimanee et al. (2014) [56] found that queens have decreased susceptibility to *N. ceranae* as they age.

Overall, pathogen loads were relatively low for many of the colonies that were screened. Co-infections with multiple pathogens were common in pooled honey bee workers, and low load may be an additional factor accounting for a lack of transmission, as ABPV, BQCV, IAPV, LSV, Trypanosome, and Nosema were present at <10^5^ genome equivalents per pooled worker sample. This was not the case for DWV worker–queen transmission, where worker samples had DWV loads >10^7^ genome equivalents per bee. Despite DWV being present at higher levels in our colonies, a comparison of DWV-negative queens that were exchanged into DWV-positive foster colonies revealed that DWV was not transmitted from worker to queen for either DWV-A or DWV-B during the 24 day exposure period. Only two queens from DWV-B-negative source colonies placed into positive foster colonies tested positive for DWV-B, and as such are the only examples of a pathogen potentially being transmitted from worker to queen. However, there is a possibility that these queens were already DWV-B-positive, as two DWV-B-positive queens from DWV-B-negative sources and foster colonies were also identified. These results suggest that DWV-positive queens acquired infection from a source other than their immediate attending workers, although our data do not enable us to speculate on the true transmission route. DWV has previously been detected in 100% (*n* = 29) of queens screened by Chen et al. (2005) [57], who subsequently went on to detect DWV in the gut and feces of queens [33], suggesting that queens can acquire infection from royal jelly or shed viral particles in the gut. DWV has been detected in every honey bee life stage [9], allowing the possibility that DWV-positive queens may acquire DWV infection vertically via their parents. Amiri et al. (2018) [58] found DWV transmission from queen to egg to be inefficient, and that high viral loads in queens were required for successful DWV transmission. Therefore, high viral load seems to be a prerequisite for DWV transmission from mother to daughter queens.

Nosema and LSV were the most abundant pathogens detected in our worker samples, a result which has also been evidenced in other studies involving US honey bee colonies [21,40,59]. Horizontal transmission routes have been identified for both Nosema and LSV. LSV has been detected in the gut, feces [59], Varroa, and pollen [60], while Nosema are suggested to be transmitted via the oral–fecal route [61]. Nosema has also been shown to be transmitted from workers to queens in in vitro bioassays [62]; however, this pattern was not evidenced in the current queen exchange experiments at the colony level. To date, LSV has not been implicated in the ongoing large-scale colony losses; however, it was detected in weak colonies that had viral loads of ≥10^8^ [59]. Given that LSV and Nosema loads did not exceed 10^6^ in the worker bees, this suggests that either loads might need to be higher to be transmitted to the queen or that queens have an alternative protective factor that is currently unaccounted.

Sampling of queens give a definite profile of the infection status of a queen within a given colony at any point in time. This is more complicated for workers, as many thousands of workers make up the “worker” population within a colony. To describe an infection status or profile of workers within a colony, pooling of workers is the most parsimonious and efficient way of sampling. It was previously reported that in a high-prevalence DWV infection scenario, a pooled sample of 30 honey bee workers provided an accurate infection measure when compared to the averaged infection levels determined from sampling individual workers from the same colony [12].

It is also important to note the levels of DWV-A and DWV-B observed in our study were approximately six-fold lower than those observed for dying colonies during the overwinter period [12,54]. This result can be explained by the low Varroa levels in the colonies sampled. However, Highfield et al. (2009) also observed that during the spring and summer months, the DWV loads within workers in any given colony may vary considerably from week to week. Workers could go from being heavily infected to having loads below limits of infection. Given the increased egg-laying of the queen over this period, it was hypothesized that the greater turnover rates of workers could explain the clearing of an infection. This was based on the inference that the queen was not passing her viruses on to her brood. Here, we can affirm that this is not only true, but that the queen does not pick up the viruses from her workers. The colony can, therefore, recover from an active virus infection during the spring and summer months, despite the workers carrying high viral loads in the weeks prior.

Viruses belonging to the acute paralysis complexes (ABPV, IAPV, KBV; [63]) have been linked to colony decline. These viruses are highly virulent and kill infected individuals in as few as three days [64,65]. Previous research using cage experiments has shown that IAPV at loads of 10^7^ is successfully transmitted to queens from workers via feeding [65]. This research also showed that queens spent equivalent periods of time with both IAPV infected and non-infected individuals, and that IAPV infection did not alter the likelihood of workers tending to the queen [65]. Therefore, transmission of ABPV and IAPV from a worker to a queen is stochastic but limited because of the reduced lifespans of infected individuals. Only one queen in our study was IAPV-positive and viral loads were low in infected workers (≤10^5^).

Alternatively, pathogen transmission may have occurred frequently from worker to queen, but the infection did not take hold, which may be explained by a number of factors. When analyzing the data as a whole, it appeared that DWV-A and DWV-B were just as likely to be detected in workers as queens. However, when analyzing the data individually we found that DWV transmission from positive foster colonies did not occur, as queens tested negative. Whilst queens were negative for the majority of pathogens detected in workers, this does not mean that these pathogens are not capable of causing infection in the queen [29,39,57,66]. Age-related immunity has been shown to protect queens from Nosema infection [56]; therefore, it is possible that over time queens build immunity for pathogens to which they are exposed. Differential gene expression between workers and queens [67] and social facilitation also contribute to the long life of the queen [44], as workers are more likely to encounter detrimental environmental stressors than the queen because of their caste-related roles. In addition, queens are fed a protein-rich diet of royal jelly and colonies fed protein-rich diets have reduced DWV viral loads [68]. Therefore, nutrition, age-related immunity, and queen physiology are all contributing factors providing protection against pathogens to queens.

In conclusion, pathogen transmission neither occurred from workers to queens for the abundant pathogens LSV and Nosema, nor did it occur for DWV-A and DWV-B, which were detected in higher loads. These findings suggest that the transmission of pathogens within a colony is not necessarily the same between adult workers and their queen, especially for adult queens newly introduced into novel nest environments. Future experiments should focus on how queens become infected with viral pathogens, especially because they are key vectors for vertical transmission within colonies.

## Figures and Tables

**Figure 1 insects-11-00382-f001:**
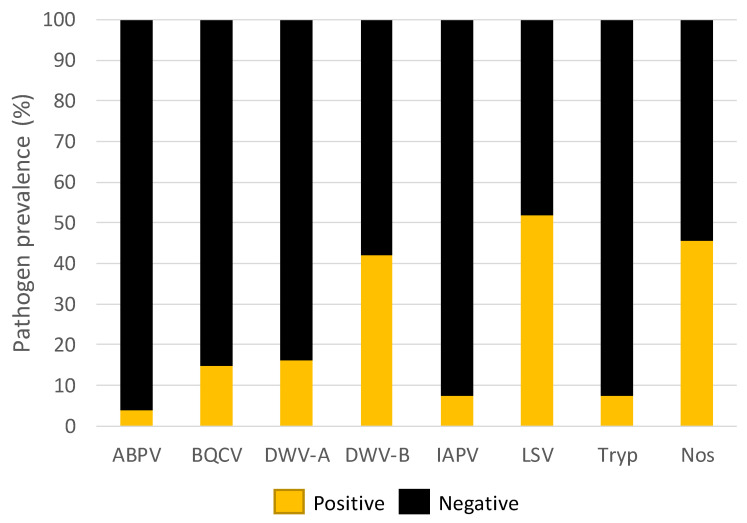
The percentage (%) of samples positive for pathogens detected within all the samples screened. Pathogens detected were Acute bee paralysis virus (ABPV), Black queen cell virus, (BQCV), Deformed wing virus-A and B (DWV-A & DWV-B), Israeli acute paralysis virus (IAPV), Lake Sinai virus (LSV), Trypanosomes (Tyrp) and Nosema (Nos).

**Figure 2 insects-11-00382-f002:**
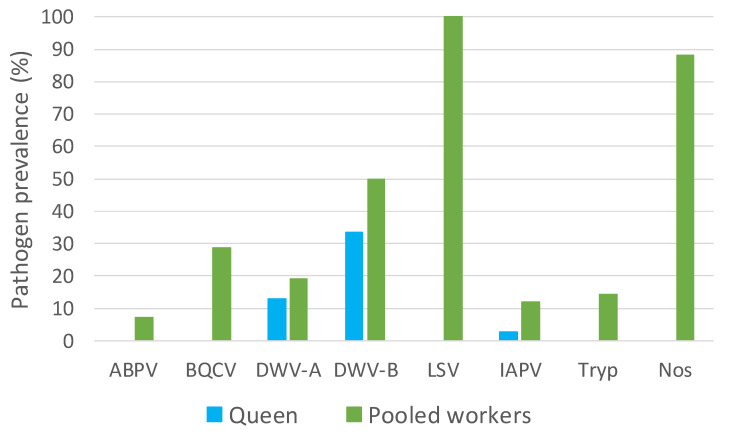
The percentages of pathogen-positive queens (blue; *n* = 39) and pooled workers (green; *n* = 42).

**Figure 3 insects-11-00382-f003:**
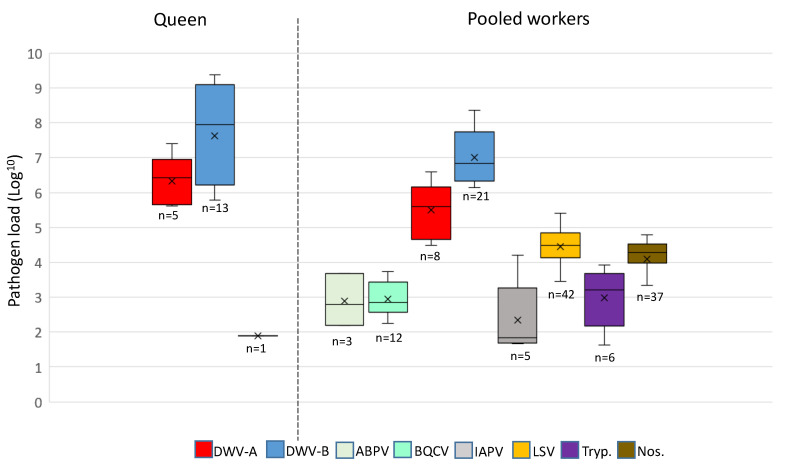
A comparison of the pathogen load (log^10^) in positive queens and pooled worker samples. Whiskers show the minimum and maximum range, horizontal lines within the boxes represent the median pathogen load, and X designates the mean pathogen load.

**Figure 4 insects-11-00382-f004:**
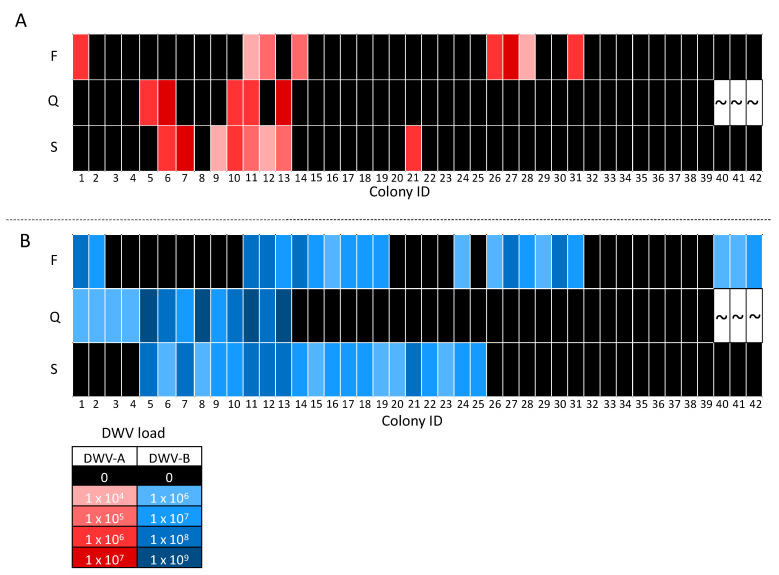
Heat map comparing DWV-A (**A**) and DWV-B (**B**) viral loads (genome equivalents) of queens and corresponding pooled worker bee samples from foster (F) and source (S) colony combinations. Colony IDs are displayed on the x axis, allowing queens to be identified by their corresponding foster and source colony. Viral loads are indicated by color intensity, whereby darker shades indicate higher viral load, black represents negative samples, and “~“ represents the three queens that died.

**Figure 5 insects-11-00382-f005:**
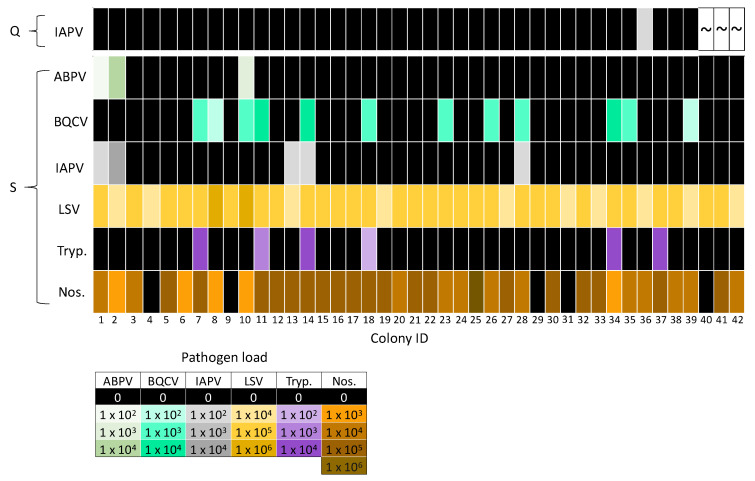
Heat map of pathogens detected in queens (Q) and pooled worker samples from source colonies (S). Queens were negative for ABPV, BQCV, LSV, Trypanosome (Tryp.), and Nosema (Nos.), so these data were not displayed. Colony IDs are displayed on the x axis. Pathogen loads are indicated by color intensity, whereby darker shades indicate higher loads, black represents negative samples, and “~” represents the three queens that died.

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
