# Peer review of "The Pathogen Profile of a Honey Bee Queen Does Not Reflect That of Her Workers"

_insects, 2020, doi:10.3390/insects11060382_

Round 1

Reviewer 1 Report

Insects Review

Notes:

The pathogen profile of a honey bee queen does not reflect that of her workers

  • Known honey bee pathogen co-occurrence
  • Pathogen transmission from workers to queens.

  • Queens from 39 colonies were removed from their source hives and introduced into a second, unrelated exchange colony.
    • Worker samples were taken from the source colony on the day of queen exchange
    • Queens collected 24 days after introduction
  • Screened for Nosema spp, Trypanosome sppe, ABPV, BQCV, CBPV, IAPV, LSV< DWVA,B,C

Results:

  • LSV, Nosema, and DWV-B were the most abundant pathogens in colonies.
  • All workers n = 39 were LSV positive,
  • 89% workers nosema positive, with low pathogen loads
  • All queens negative for both LSV and Nosema
  • No evidence of DWV transmission from worker to queen

Honey bee pathogen presence and diversity in queens cannot be revealed from screening workers, nor were pathogens successfully transmitted to the queen.

Comments:

This paper is important because queen issues are a current top complaint/ issue for beekeepers. Because the queen outlasts the workers, it would have been very helpful to have an assay that would predict queen health based on workers screening.  While this paper demonstrates that a straight forward relationship is unlikely, it presents valuable information on queen health that can be used to better understand colony health.

Hypothesis: If queens are constantly exposed to pathogens within the colony by attending worker bees and these pathogens are transmitted to the queen, then the pathogen profiles of the queens will match that of the workers.

Minor comments:

-Line 106: ‘were selected based on their colony phenotype of sealed brood pattern” – This is confusing for a few reasons:

  • This line and the referenced paper do not indicate what colonies were chosen. Please explain how brood pattern was used to select colonies (i.e. Were they all 5s? 3s?).
  • It is not clear why brood pattern is used when the referenced paper states that brood pattern is not associated with queen pathogen presence.
  • Figure 2 caption:
    • The way that the caption reads is deceptive, because it reads like there were 39 positive detections for queens and 39 positives for workers. However, that is not what is represented by the blue lines.  If we look at figure 3, it is a maximum of N = 19 queens (max – we can’t tell because co-infections aren’t reported).
    • Also, the N= 39 at the top left looks tacky, and isn’t accurate (the total isn’t 39).
    • Also I would use “pooled workers” instead of worker so it is clear that this isn’t an actual prevalence in workers.
  • Lines 227 is repeated in line 264-265 (I think it is better at line 265)

Grammar –

  • Line 55 – misuse of colon
  • Line “preforming” should be “performing”

Medium comments

The biggest issue with the way the results are presented is that they do not match the title.  I was really excited to read this paper, because the premise of matching the infections in the queen to the infections in the hive is really important and interesting to beekeepers.  However, never in the paper do we actually learn the pathogen profiles of each sample set, nor do we get a description of the relationship of the pathogens found in queens to the pathogens found in the source or exchange colonies.  I get that the bulk of the infections weren’t present in queens, but why aren’t these data presented for the three pathogens that were found in queens?

  • For the queen with IAPV – was she in a hive with IAPV at the beginning, the end or never?
  • For the queens with DWV A and B – what was the likelihood that she came from a colony that had DWV (either source or exchange)? This is touched on briefly in line 205 (source colonies), but should be fully discussed.  How many were in positive exchange colonies? Were there any where the queen had DWV and the bees didn’t?  Did it matter by DWV A or DWVB?   Can we predict at all the likelihood that a queen will have DWV A or B if one of the colonies she was in had it? I had trouble with figure 4 / section 3.3, so it may be that I missed this, couldn’t pull this info out (see comments below).
  • Section 3.1 – It would be useful to know what % of queens and colonies had single infections vs no infections vs co-infections (the pathogen profile). Right now, it is impossible to tell if any queens or worker groups were free from tested pathogens.  Please report on co-infections and hives that did not have infection – even if just descriptively. 

  • Line 170 – “Workers were more likely to be positive for pathogens compared to queens’
    • ‘likely’ is a difficult word – it indicates that you compared the likelihood of the event occurring, which is not what was done. It was a comparison of relatedness of distributions of individual queens versus distributions of pooled workers.
    • It is a bit problematic to compare prevalence between workers and queens because the workers are from pooled data while the queens are individually. This would not be as big as a problem if all the diseases were common, but since at least three are classified as rare, there is a chance that queens are equally likely as workers in becoming infected. It would have been awesome if there were data on individual workers (to evaluate if the variability within the hive), but since this was not done, I don’t know of any analysis off hand (but I’m not a statistician).  This may just be remedied with a more careful word choice, more care in statements, and a disclaimer in the discussion. 
  • Line 184 – Please indicate if this is in workers, queens, or all samples.
  • Figure 4 / section 3.3 - I re-read this section a bunch, and I am still confused about this. I think that it is just a wording/ term issue, but it is really a struggle to understand.
    • I feel pretty comfortable with the term sample set being the three samples that are linked by the 39 queens (Source hive, queen, exchange hive).
    • I can see clearly from figure 3 that N = 18 queens had DWV (DWV-A N=5, DWV-B N =13). However, in line 202 you state that there are 46 ‘positives’ – but it is unclear what a positive.  I originally thought that meant either a queen or worker pool that tested positive for either DWV, but that number is 44 if you take the N from figure 3 (5+13+7+9). What is a positive that would not be in figure 3?  What is your reader missing here?  
    • Intuitively, I can only see how there would be 39 columns in figure 4, one representing each sample set. I don’t understand what each sample ID is in reference to.
    • In figure 3, you show the pathogen load of DWV – A and B in queens. However in figure 4, you only show DWV- B in queens.  Why isn’t DWVA shown for queens in this figure? 
  • I think that this could be remedied with a rewording of the line on 202, a better description of positive in the context of your sample sets, and a clearer, more detailed caption for figure 4.

Author Response

Reviewer one  

Minor comments:

Line 106: ‘were selected based on their colony phenotype of sealed brood pattern” – This is confusing for a few reasons:

  • This line and the referenced paper do not indicate what colonies were chosen. Please explain how brood pattern was used to select colonies (i.e. Were they all 5s? 3s?).
  • It is not clear why brood pattern is used when the referenced paper states that brood pattern is not associated with queen pathogen presence.---

- Thank you for both these points, this was in deed confusing. The brood laying pattern was not correlated with queen health status. Consequently, we have updated the text to read -  Line 105 ‘All colonies were headed by queens < 6 months old and were not selected based on their sealed brood pattern as part of the Lee et al., (2019) [49] study.  No association between brood pattern and pathogen presence was made by Lee et al., (2019); however, the samples provide the opportunity to assess intra-colony pathogen transmission.

  • Figure 2 caption:

The way that the caption reads is deceptive, because it reads like there were 39 positive detections for queens and 39 positives for workers. However, that is not what is represented by the blue lines.  If we look at figure 3, it is a maximum of N = 19 queens (max – we can’t tell because co-infections aren’t reported).

- Good point.

  • We removed the n=39 insert.
  • We also presented the percentage of positives in figure 2 and then the viral load of these positives in figure 3.
  • Co-infection with DWV-A and DWV-B occurred in queens and respective source & foster colonies can now been seen in figure 4 by tracking colony ID.
  • Updated section “3.3 DWV transmission between pooled workers and queens” accordingly.
  • Added figure 5 and 6, showing co-infection in worker samples.

  • Also, the N= 39 at the top left looks tacky, and isn’t accurate (the total isn’t 39).

– Removed.

  • Also I would use “pooled workers” instead of worker so it is clear that this isn’t an actual prevalence in workers.

– Added the word ‘Pooled’ thought MS.

  • Lines 227 is repeated in line 264-265 (I think it is better at line 265)

– We have changed the text

  •  
  • Grammar –
  • Line 55 – misuse of colon

– Corrected.

  • Line “preforming” should be “performing”

– Corrected.

Medium comments

The biggest issue with the way the results are presented is that they do not match the title.  I was really excited to read this paper, because the premise of matching the infections in the queen to the infections in the hive is really important and interesting to beekeepers.  However, never in the paper do we actually learn the pathogen profiles of each sample set, nor do we get a description of the relationship of the pathogens found in queens to the pathogens found in the source or exchange colonies. I get that the bulk of the infections weren’t present in queens, but why aren’t these data presented for the three pathogens that were found in queens?

- Thank you for these helpful comments. We have added additional information at the colony level so that within colony and co-infection correlates can be made. As mentioned above, additional figures and text were added.

  • For the queen with IAPV – was she in a hive with IAPV at the beginning, the end or never?

– The source and exchange colony were not IAPV positive, there was no relationship – Text added to line 237.

  • For the queens with DWV A and B – what was the likelihood that she came from a colony that had DWV (either source or exchange)? This is touched on briefly in line 205 (source colonies), but should be fully discussed.  How many were in positive exchange colonies? Were there any where the queen had DWV and the bees didn’t?  Did it matter by DWV A or DWVB?   Can we predict at all the likelihood that a queen will have DWV A or B if one of the colonies she was in had it? I had trouble with figure 4 / section 3.3, so it may be that I missed this, couldn’t pull this info out (see comments below).

– As stated above, Section 3.3 has now been completely revised and these points have been addressed.

  • Section 3.1 – It would be useful to know what % of queens and colonies had single infections vs no infections vs co-infections (the pathogen profile). Right now, it is impossible to tell if any queens or worker groups were free from tested pathogens.  Please report on co-infections and hives that did not have infection – even if just descriptively.

–Thank you, we have revised section 3.3 and added figure 6 that shows the number of pathogens detected and how many colonies these were detected in. The text has been modified.

  • Line 170 – “Workers were more likely to be positive for pathogens compared to queens ’‘likely’ is a difficult word – it indicates that you compared the likelihood of the event occurring, which is not what was done. It was a comparison of relatedness of distributions of individual queens versus distributions of pooled workers.

– Thank you. We revised throughout to give exact numbers and percentages observed.

  • It is a bit problematic to compare prevalence between workers and queens because the workers are from pooled data while the queens are individually.

We respectfully disagree. We are framing our experiment in the context of a colony, not individuals. Therefore, in a colony it is often only one queen exist at any one time.  This is naturally not the case for workers. We nonetheless added additional text to the Discussion expanding on this exact point (lines 302 to 338.)

  • This would not be as big as a problem if all the diseases were common, but since at least three are classified as rare, there is a chance that queens are equally likely as workers in becoming infected. It would have been awesome if there were data on individual workers (to evaluate if the variability within the hive), but since this was not done, I don’t know of any analysis off hand (but I’m not a statistician).  This may just be remedied with a more careful word choice, more care in statements, and a disclaimer in the discussion. 

-The expense of carrying out an individual based honey bee study is not justifiable. In addition, pooling workers provides a good cost-effective way of carrying out an intra- and inter-colony prevalence study. In our study DWV was detected in the queens and workers when compared to the other pathogens. We are not claiming that queens cannot become infected, but suggest that the queens are not showing the same pathogen profile as the workers.

  • Line 184 – Please indicate if this is in workers, queens, or all samples.

- All samples are now shown

  • Figure 4 / section 3.3 - I re-read this section a bunch, and I am still confused about this. I think that it is just a wording/ term issue, but it is really a struggle to understand.

–Figure 4 & section 3.3. has now been extensively revised.

  • I feel pretty comfortable with the term sample set being the three samples that are linked by the 39 queens (Source hive, queen, exchange hive).
  • I can see clearly from figure 3 that N = 18 queens had DWV (DWV-A N=5, DWV-B N =13). However, in line 202 you state that there are 46 ‘positives’ – but it is unclear what a positive.  I originally thought that meant either a queen or worker pool that tested positive for either DWV, but that number is 44 if you take the N from figure 3 (5+13+7+9). What is a positive that would not be in figure 3?  What is your reader missing here?  

- Apologies, we changed the figures and updated text in the methods. 

  • Intuitively, I can only see how there would be 39 columns in figure 4, one representing each sample set. I don’t understand what each sample ID is in reference to

– We revised Figure 4, as well as the figure legend.  

  • In figure 3, you show the pathogen load of DWV – A and B in queens. However in figure 4, you only show DWV- B in queens.  Why isn’t DWVA shown for queens in this figure? 

- We apologize, this was an oversight and the figures have been updated. 

  • I think that this could be remedied with a rewording of the line on 202, a better description of positive in the context of your sample sets, and a clearer, more detailed caption for figure 4.

- corrected

Reviewer 2 Report

This study investigated transmission of pathogens, in particular viruses, between the honeybee workers  and the queen. The experimental involved 39 colonies between which queens were exchanged. Samples of the worker bees were collected from each colony at the time of the queen exchange and the queens were sampled 24 days later to allow acquisition of the pathogens from “exchange” colonies.  Each of 78 samples (39 queens + 39 worker pools) were tested for the presence of six honeybee viruses, as well as Nosema and Trypanosome spp by RT-qPCR and the pathogen profiles of the queens and the adult worker bees from their respective “source” and “exchange” colonies were compared. The study showed that the queen pathogen profiles did not match those of the workers in their respective  “source” or “exchange” colonies. Only DWV-A and DWV-B were detected in the queens. Also, LSV and Nosema, which were present at low levels in almost all worker bees, were not detected in the queens.

The results of this study are novel and contribute to further understanding of maintenance and circulation of honey bee pathogens  and therefore will be of interest to entomologists and bee pathologists, but several points should be addressed. Importantly, the paper must revise Figure 4 and/or Materials and Methods to resolve apparent contradictions between Figs.  2,3  and Fig. 4 described below. 

L104-115 and Figure 4. According to Materials and methods each colony involved in the study acted as “source” for one queen and as “exchange” for another queen.  The line 115 states that “ …This resulted in 78 samples, a total 39 queens and 39 pooled samples of 50 workers“. If it is a case, why did Figure 4 showed 46 sets of queens, “source”: and “exchanged” samples? Also, if the same set of 39 colonies acted as “source” and “exchange” there must be the same number of worker bee samples with the same virus sets, but Figure 4 suggest that there are 4 source (S)  colonies with DWV-A (marked pink to read)  and 7 exchange (E) colonies with  DWV-A  (marked pink to read). 

L112. Specify the age of the sampled worker bees. Did the worker samples contained exclusively in-hive (nurse) bees or included both nurse bees and foragers? 

Lines 104-105. Specify Varroa infestation levels of the colonies involved. This could be relevant to the levels of Varroa -vectored DWV-A and DWV-B. 

Lines 145-146. “Pathogen prevalence” . Provide detection thresholds for the pathogens.  

Lines 163-165 and Figure 2. Provide 95% confidence intervals for prevalence of each detected pathogen. 

Figure 2. Specify which pathogen pathogen was present in the queen samples (n=1) in addition to DWV-A and DWV-B . 

Line 174 “DWV-A was present in 19% of workers and 12% of queens” & Figures 2, 3, 4. 

There is contradiction between ~12% prevalence of DWV-A reported in Line 174 and Figs 2, 3 and Figure 4 which apparently shows queen have no DWV-A at all. 

According to Materials and methods (L104), Figures. 1 and 2,  there were 39 queens and colony 39 worker samples (each of them served as “source” and “exchange” for any given queen. Figure 4 showed 46 samples. Revise the Figure 4 or correct Materials and methods.

Figure 4. "DWV-A and DWV-B." There is a contradiction between Figure 4 which showed that only DWV-B is present in the queens and Figure 3 which showed that DWV-A is also present in the queens (above 10^7 copies in some of them). Figure 4 should be redesigned to include all DWV-A and DWV-B data for example by making two rows for each sample type - for DWV-A and for DWV-B.

Author Response

Reviewer two.

This study investigated transmission of pathogens, in particular viruses, between the honeybee workers  and the queen. The experimental involved 39 colonies between which queens were exchanged. Samples of the worker bees were collected from each colony at the time of the queen exchange and the queens were sampled 24 days later to allow acquisition of the pathogens from “exchange” colonies.  Each of 78 samples (39 queens + 39 worker pools) were tested for the presence of six honeybee viruses, as well as Nosema and Trypanosome spp by RT-qPCR and the pathogen profiles of the queens and the adult worker bees from their respective “source” and “exchange” colonies were compared. The study showed that the queen pathogen profiles did not match those of the workers in their respective  “source” or “exchange” colonies. Only DWV-A and DWV-B were detected in the queens. Also, LSV and Nosema, which were present at low levels in almost all worker bees, were not detected in the queens.

The results of this study are novel and contribute to further understanding of maintenance and circulation of honey bee pathogens  and therefore will be of interest to entomologists and bee pathologists, but several points should be addressed. Importantly, the paper must revise Figure 4 and/or Materials and Methods to resolve apparent contradictions between Figs.  2,3  and Fig. 4 described below. 

L104-115 and Figure 4. According to Materials and methods each colony involved in the study acted as “source” for one queen and as “exchange” for another queen.  The line 115 states that “ …This resulted in 78 samples, a total 39 queens and 39 pooled samples of 50 workers“. If it is a case, why did Figure 4 showed 46 sets of queens, “source”: and “exchanged” samples? Also, if the same set of 39 colonies acted as “source” and “exchange” there must be the same number of worker bee samples with the same virus sets, but Figure 4 suggest that there are 4 source (S)  colonies with DWV-A (marked pink to read)  and 7 exchange (E) colonies with  DWV-A  (marked pink to read). 

– Thank you, the text and figures have now been revised so it is clearer how the samples were collected.

L112. Specify the age of the sampled worker bees. Did the worker samples contained exclusively in-hive (nurse) bees or included both nurse bees and foragers?

- Agreed, the added additional to line 114.

Lines 104-105. Specify Varroa infestation levels of the colonies involved. This could be relevant to the levels of Varroa -vectored DWV-A and DWV-B. 

– Thank you, text has been added to line 124. Colonies were low for Varroa; the transmission of pathogens vectored by Varroa was not considered, as we were interested in transmission from workers to queens.

Lines 145-146. “Pathogen prevalence” . Provide detection thresholds for the pathogens. 

- Updated and note new figure 5.

Lines 163-165 and Figure 2. Provide 95% confidence intervals for prevalence of each detected pathogen.

- This is not the average pathogen prevalence, this figure shows the percentage of queens or workers that were pathogen positive. 95% CI are not applicable here because it is percentage data.

Figure 2. Specify which pathogen pathogen was present in the queen samples (n=1) in addition to DWV-A and DWV-B

-The one queen was IAPV positive, this is mentioned in the text – Line 199. We have also added a Figure 5, that shows co-infection in workers and the one IAPV positive queen. Co-infections are discussed in the results section 3.3, which has been completely revised.

Line 174 “DWV-A was present in 19% of workers and 12% of queens” & Figures 2, 3, 4. 

There is contradiction between ~12% prevalence of DWV-A reported in Line 174 and Figs 2, 3 and Figure 4 which apparently shows queen have no DWV-A at all. 

– Thank you and we apologize for this oversight, we have now amended the figures.

According to Materials and methods (L104), Figures. 1 and 2, there were 39 queens and colony 39 worker samples (each of them served as “source” and “exchange” for any given queen. Figure 4 showed 46 samples. Revise the Figure 4 or correct Materials and methods.

– Thank you, we have revised the methods to clarify this Line 104 onwards. We have revised figure 4.

Figure 4. "DWV-A and DWV-B." There is a contradiction between Figure 4 which showed that only DWV-B is present in the queens and Figure 3 which showed that DWV-A is also present in the queens (above 10^7 copies in some of them). Figure 4 should be redesigned to include all DWV-A and DWV-B data for example by making two rows for each sample type - for DWV-A and 

- Resolved.

Reviewer 3 Report

The manuscript " The pathogen profile of a honey bee queen does not reflect that of her workers ", presents a comparison between pathogen profile of queens and workers within the same colony with attempt mainly to evaluate the horizontal transmission between the queen and her respective workers using fostering approach: when queens are moved from their original colonies to a new one. The workers and queen are compared for their pathogenic load. The results clearly show the differences between the worker and queen's in loads of most pathogens and thus in ability to predict the health status of queens based on workers analysis.

The issue of such a transmission is an important one from both scientific prospect and even more so for a practical apiculture as queens are mass produced and transported within and between countries.

The manuscript is written in general a very clear form, the research question is well defined even thou not very original. However unlike previous studies, this one evaluates the viral transmission in a real apicultural conditions. The researchers use a large sample size of 39 colonies that grants high validity to their findings. The experimental design is simple but smart and the results are quite surprising.

My problem is that the described protocol a bit confusing and unfortunately did not last longer as it would be very interesting to analyze also the daughters of the queens.

The collection of workers is described twice, it seems as if two types of workers samples were collected: one from the original colonies and another from the foster colonies. But in this case you would expect to have 78 samples of workers and 39 of queens, but latter this is not the case. In figure 4 there are 138 samples. Where are these samples are coming from? This needs to be clarified. I would suggest to rewrite and unite the two paragraphs of section 2.1.

Moreover, in figures 2 to 3 there were 39 groups of workers, and from the legends and the text it is not clear what kind of workers were they.

It is also not clear to me why the researchers waited 24 day to collect the queens from the foster colony?

I have also some reservations concerning the methodology for Nosema detection used in this study. As far as I know unlike viruses, Nosema infection is not evaluated by RNA copies but its DNA. A reference for such an approach would help.

I wonder what were exactly the criteria for colony selection. Was health condition of the colony considered as one of the parameters? If so, how?

Some specific comments are listed below:

Suggest to change "exchange colony" to foster colony

  1. 200-201 something is confusing here: there suddenly two groups of workers.

And in Fig 4. There are 46 X 3 samples, this is really confusing when there were just 78 at the beginning.

Author Response

Reviewer three

The manuscript " The pathogen profile of a honey bee queen does not reflect that of her workers ", presents a comparison between pathogen profile of queens and workers within the same colony with attempt mainly to evaluate the horizontal transmission between the queen and her respective workers using fostering approach: when queens are moved from their original colonies to a new one. The workers and queen are compared for their pathogenic load. The results clearly show the differences between the worker and queen's in loads of most pathogens and thus in ability to predict the health status of queens based on workers analysis.

The issue of such a transmission is an important one from both scientific prospect and even more so for a practical apiculture as queens are mass produced and transported within and between countries.

The manuscript is written in general a very clear form, the research question is well defined even thou not very original. However unlike previous studies, this one evaluates the viral transmission in a real apicultural conditions. The researchers use a large sample size of 39 colonies that grants high validity to their findings. The experimental design is simple but smart and the results are quite surprising.

- Thank you

My problem is that the described protocol a bit confusing and unfortunately did not last longer as it would be very interesting to analyze also the daughters of the queens.

- This was an opportunistic study conducted from an earlier experimental design.  Future work will certainly look at the brood and daughters of the queen.

The collection of workers is described twice, it seems as if two types of workers samples were collected: one from the original colonies and another from the foster colonies. But in this case you would expect to have 78 samples of workers and 39 of queens, but latter this is not the case. In figure 4 there are 138 samples. Where are these samples are coming from? This needs to be clarified. I would suggest to rewrite and unite the two paragraphs of section 2.1. 

- We apologize that the description of sampling was confusing and we have clarified the text. – line 115 onwards.

Moreover, in figures 2 to 3 there were 39 groups of workers, and from the legends and the text it is not clear what kind of workers were they.

– Thank you, figures 2 and 3 show the data as a whole. Workers were not separated into source or foster colony in the figures. Text has been updated.

It is also not clear to me why the researchers waited 24 day to collect the queens from the foster colony?

-Additional text has been added to line 116.

I have also some reservations concerning the methodology for Nosema detection used in this study. As far as I know unlike viruses, Nosema infection is not evaluated by RNA copies but its DNA. A reference for such an approach would help.

– Sorry for the confusion, the text has been improved -  line 139.

I wonder what were exactly the criteria for colony selection. Was health condition of the colony considered as one of the parameters? If so, how?

-  Thank you, we have added text to text to line 107.

Some specific comments are listed below:

Suggest to change "exchange colony" to foster colony

– Thank you for this suggestion, it has vastly improved the text, we have updated this throughout MS.

200-201 something is confusing here: there suddenly two groups of workers. And in Fig 4. There are 46 X 3 samples, this is really confusing when there were just 78 at the beginning.

-Thank you, we have improved the text regarding sampling from line 115 onwards. We have also revised and updated figure 4, and have improved the figure legend.

Round 2

Reviewer 3 Report

I  found the revised manuscript very clear.

I have just very minor comments.

Listed bellow.

The results clearly show that there is no pathogen transmission from workers to queens and visa versa. However, these  results were acquired in colonies with very low Varroa infection if at all. I wonder if it would be the same under Varroa infestation that has effect both on the viruses pathogenicity and honey bee immunity. I think that it will be good to address this issue shortly in the discussion.  

Fig 5: delete "co" from co infection in Q as only infection in queens is shown.

Fig 6: is not rely needed as the result is described in details in the text.

Author Response

Listed below.

The results clearly show that there is no pathogen transmission from workers to queens and visa versa. However, these  results were acquired in colonies with very low Varroa infection if at all. I wonder if it would be the same under Varroa infestation that has effect both on the viruses pathogenicity and honey bee immunity. I think that it will be good to address this issue shortly in the discussion.  –

Thank you. A sentence has been added to line 307.

Fig 5: delete "co" from co infection in Q as only infection in queens is shown.

  • Done

Fig 6: is not rely needed as the result is described in details in the text.

  • Done
